# The Melanoma-Associated Antigen Family A (MAGE-A): A Promising Target for Cancer Immunotherapy?

**DOI:** 10.3390/cancers15061779

**Published:** 2023-03-15

**Authors:** Alaa Alsalloum, Julia A. Shevchenko, Sergey Sennikov

**Affiliations:** 1Laboratory of Molecular Immunology, Federal State Budgetary Scientific Institution Research Institute of Fundamental and Clinical Immunology, 630099 Novosibirsk, Russia; 2Faculty of Natural Sciences, Novosibirsk State University, 630090 Novosibirsk, Russia; 3Department of Immunology, V. Zelman Institute for Medicine and Psychology, Novosibirsk State University, 630090 Novosibirsk, Russia

**Keywords:** cancer-testicular antigen, MAGE-A, cancer immunotherapy, adoptive T-cell therapy, cancer vaccine

## Abstract

**Simple Summary:**

Given that selecting an appropriate target is the first step in developing effective cancer immunotherapy, we focus on melanoma-associated antigens (MAGEs), which are a subclass of cancer/testis (CT) antigens characterized by restricted expression in immune-privileged tissues and a variety of cancers. For a long time, possessing this combination of characteristics has made MAGEs a remarkable candidate for effective treatment. Based on promising clinical data from combinatorial therapies, ongoing clinical trials targeting MAGE-A antigens, as well as a thorough understanding of the crosstalk between cancer and immune control, a new era in cancer immunotherapy is expected. The goal of this review is to provide an in-depth understanding of the various strategies for targeting the MAGE-A family for cancer treatment, as well as to highlight some of the most exciting recent advances in this area.

**Abstract:**

Early efforts to identify tumor-associated antigens over the last decade have provided unique cancer epitopes for targeted cancer therapy. MAGE-A proteins are a subclass of cancer/testis (CT) antigens that are presented on the cell surface by MHC class I molecules as an immune-privileged site. This is due to their restricted expression to germline cells and a wide range of cancers, where they are associated with resistance to chemotherapy, metastasis, and cancer cells with an increasing potential for survival. This makes them an appealing candidate target for designing an effective and specific immunotherapy, thereby suggesting that targeting oncogenic MAGE-As with cancer vaccination, adoptive T-cell transfer, or a combination of therapies would be promising. In this review, we summarize and discuss previous and ongoing (pre-)clinical studies that target these antigens, while bearing in mind the benefits and drawbacks of various therapeutic strategies, in order to speculate on future directions for MAGE-A-specific immunotherapies.

## 1. Introduction

Our immune system is a complex network of different cells and proteins that function together in a coordinated way. If we imagine our body to be a homeland, the immune system can be viewed as the military, which protects us from invaders and threats. Just as the military is divided into different forces, the immune system is also split into specialized branches. All cooperate with each other to keep the body safe from injury. B and T cells are among the immune system’s forces and circulate in the blood, lymphoid tissues, and at effector sites to immediately respond to any harm. For example, T cells directly attack and kill foreign bodies. B cells secrete proteins called antibodies, which can flag the foreign agent for elimination. The immune system is regularly scanning for mutations in the cell that could lead to cancer. The generation of new forms of host molecules, known as tumor-specific antigens (TSA) or tumor-associated antigens (TAA), characterizes cancer cells and allows the immune system to recognize cancer tissue [1].

However, these spontaneous immune responses in cancer patients fail to control tumor growth because cancer cells are genetically unstable, which can enable them to develop mechanisms to escape from the immune system, including the concealment of cancer antigens or secretion of immune-suppressing molecules. Consequently, significant research efforts to combat cancer using a combination of surgical, chemical, and radiation therapy have been initiated. However, most of the available anticancer drugs are non-specific and lead to side effects [2].

Recent advancements have paved the way for immunotherapy to be recognized as a promising tool that can react or reboot the immune system to attack again and destroy the cancer cell with fewer side effects compared with traditional cancer treatment. This can be achieved in various ways, such as with monoclonal antibodies, adoptive T cell therapy, and therapeutic vaccines, which represent major steps toward targeted therapy. One of the most difficult challenges, however, is designing a therapy that is both specific and effective for cancer. Selecting an ideal target is the first step in developing antigen-specific cancer immunotherapy that is efficacious. It should be stably expressed on the cancer cell, disrupting the mechanisms of tumor escape from immunity. Antigens must be highly restricted in expression on healthy cells and thus protect against autoimmunity [3].

Cancer/testis (CT) antigens match these criteria for an ideal target, as CT antigens are tumor antigens that are identified with a normal expression and restricted to testicular, ovarian, fetal, and placental germ cells [4,5], but may also be detected in malignancies [6,7]. CT antigens are normally located in an immune-privileged site due to their expression in the isolated immune environment of the seminiferous epithelium, which is protected from immune cells by the so-called blood–testicular barrier. In addition, class I MHC molecules are absent from germ cells [8].

MZ2-E (now known as MAGE-A1), the first cancer/testis antigen, was discovered nearly 31 years ago in 1991. It was discovered via the recognition of a melanoma cell line in vitro by autologous cytotoxic CD8 T cells. MAGE-A1 emerged as an antigen presented on the cell surface by the major histocompatibility complex (MHC) [9]. Shortly after this discovery, more proteins with the same properties were identified. Over 60 proteins that belong to this family are further subdivided into two groups based on the location of their genes [10]. Type I, whose expression is restricted to the X chromosome, comprises the three subfamilies MAGE-A, -B, and -C, whereas type II, which is not limited to the X-chromosome, comprises MAGE-D, -E, -F, -G, -H, -L, and Necdin [11].

Both groups, type I and type II, have a MAGE homology domain (MHD) that is around 170 amino acids long and highly conserved within the MAGE-A subfamily (>80% identical) [12]. Structural studies have shown that the MHD is a tandem-winged helix motif that was assumed to play a role in protein–protein interactions [13]. Several studies were then conducted to better understand how relatively similar MAGE proteins can have distinct functions. It was proposed that the adaptable MHD undergo allosteric modifications, thus allowing interactions with different protein domains and conferring special properties to MAGE members [14,15,16].

MAGEs have been found to be highly expressed in the male germ line and in placental cells as well as in various tumor types, including melanoma [17], colon [18], breast [19], ovarian [20], and lung [21], among others.

### MAGE-A Antigens and Cancer

Tumor cells, according to certain studies, tend to co-express two or more MAGE-A antigens (MAGE-As) at random; for example, MAGE-A3 and -A9 expression was observed in non-small cell lung cancers and is significantly correlated with poor patient survival [22,23]. In addition, other studies have shown that MAGE-A is expressed mainly in malignant tumors or metastases. In hepatocellular carcinoma, MAGE-A9 expression is significantly correlated with metastasis, tumor progression, and portal vein invasion [24]. Moreover, additional studies support the contribution of MAGE-As to oncogenesis or tumorigenesis by direct interaction with the p53 tumor suppressor. MAGE-A2, for example, can bind to the p53 core and downregulate its transactivation function [25].

Other studies have found that MAGE-As indirectly increase the survival of cancer cells through their interaction with other proteins as adaptors, especially E3 ubiquitin [26]. The helical region MHD of MAGE-A2, MAGE-A6, and MAGE-A3 was discovered to interact with the TRIM28 RING E3 ubiquitin ligase, enhancing ubiquitin ligase activity by binding to and marking p53 substrates for ubiquitination by the E3 ligase complex, thus resulting in their degradation in a proteasome-dependent manner (Figure 1A) [13].

In addition, the mechanisms that control the aberrant re-expression of MAGE-As have been suggested, at least in part. It was shown that using inhibitors of DNA methyltransferases (DNMTs), such as 5-aza-2′-deoxycytidine (decitabine; DAC), led to increased expression of MAGE-A1 in different cancer cell lines [27]. In addition, this effect can be reinforced through use of histone deacetylase inhibitor trichostatin [28]. This helps to explain why MAGE-As are rarely expressed in somatic tissues: the DNA hypermethylation of CpG dinucleotides in promoters prevents transcription factors from binding, thus inhibiting expression of MAGE-A genes [29,30]. The restricted expression of MAGE-A antigens, combined with their oncogenic activity, have received continued interest, and been investigated over time as worthy targets of therapeutic cancer strategies.

In this review, we look at past and current (pre-)clinical studies that have been conducted to target the MAGE-A family in various therapeutic approaches, with an eye toward possible advancements in MAGE-A-specific immunotherapy.

## 2. Cancer-Fighting MAGE-A-Targeting Strategies

Our current understanding, at least in part, of the mechanisms of the MAGE-A family member proteins’ expression in cancer cells suggests that it would be useful to select antigens whose loss would reduce the capability of cancer cells to thrive. There are three potential approaches for targeting MAGE-As in cancer treatment: (1) interrupting MAGE-A partner interaction; (2) targeting the regulatory pathways responsible for functional expression; and (3) immunotherapy against MAGE-As.

### 2.1. Inhibitors of MAGE-A/Partner Interaction

Recent advances in the molecular biology of MAGE-A proteins have made it possible to modify protein interactions as an adjunct to conventional cancer therapy. This strategy could have only a slight impact on normal tissues and thus minimize possible side effects due to the highly limited expression of MAGE-As in somatic tissue. There is now growing evidence that chemical compounds could disrupt binding between MAGE-A3 and TRIM28 proteins (complexed with a ubiquitin E3 ligase) based on the screening of a library of small molecules that selectively cause the death of MAGE-A-positive cells, activate p53, and reduce caspase activity. Results of the screening revealed three molecules that interfere with MAGE and TRIM28 binding and may be used as novel therapeutic agents if they do not overlap with the main homeostatic functions of TRIM28 (Figure 1B) [31].

### 2.2. Targeting the Regulatory Pathways Responsible for Functional Expression

MAGE-A expression has been shown to be correlated with DNA demethylation in various tumors [32]. It has also been observed to be inducible by demethylating agents, such as chemotherapeutic compounds that are commonly used in cancer treatment, such as decitabine [33]. Accordingly, targeting the regulatory pathways of MAGE-As may indirectly impact their expression, as it may also affect other molecular pathways. Thus, although MAGE-As cannot be selectively modified using this strategy, it could benefit in combination with other approaches that concretely target MAGE-As. Each of these specific potential methods are discussed below.

### 2.3. Immunotherapy against MAGE-As

The immune-privileged nature and oncogenic activity of MAGE-As make them universal antigens that have the capacity to elicit immune responses that are highly specific for cancer and disrupt immune tolerance in the tumor microenvironment. Interestingly, a key finding was that tumor MAGE-A10-specific T cells are common in hepatocellular carcinoma [34]. Importantly, MAGE-As gene expression has been observed in up to 70% of HCC cases [35]. Thus, there is a correlation between MAGE-As antigen expression and the cytolytic activity of tumor immune cell infiltrates. Therefore, significant effort has been initiated to develop cancer vaccines targeting MAGE-As.

#### 2.3.1. MAGE-As Antigen Vaccination

Cancer vaccines are a type of active immunotherapy that supports the patient’s immune system in its battle against cancer. Many models of cancer therapeutic vaccines have been designed, such as protein or peptide vaccines, vector vaccines, DNA or RNA vaccines, and cell-based vaccines (Figure 2A) [36]. Most clinical trials evaluating MAGE-A protein or peptide-based vaccines are focused on administering recombinant MAGE-A protein in conjunction with immunostimulatory agents to strengthen and sustain immune responses. For example, in a randomized phase II study, 75 patients with stage III or IV M1a melanoma were given recombinant protein (recMAGE-A3) in combination with either AS02B or AS15 immunostimulants to induce tumor-specific immune responses. Humoral and cellular responses to MAGE-A3 were observed in all patients (Figure 2B), but three complete responses were observed in the group that was treated with MAGE-A3 protein, which was combined with an AS15 immunostimulant (NCT00086866) [37].

According to the preliminary data, in larger phase III trials known as DERMA, the AS15 immunostimulant was combined with the MAGE-A3 protein. There was no improvement in terms of disease-free survival when compared with a placebo in 1345 patients with stage IIIB or IIIC melanoma; thus, the trial was canceled in 2015 [38]. In the settings of this clinical trial, the single antigen-specific cancer vaccine did not demonstrate sufficient efficacy. Based on these findings, the same MAGE-A3 antigen immunotherapeutic treatment was co-administered with chemotherapy (dacarbazine) in an open phase I/II study on patients with metastatic melanoma. One complete response was reported. Although only a slight clinical benefit was observed in that study, it provided interesting insights into combination therapies (NCT00849875) [39].

In another randomized, double-blind, phase III trial in patients with non-small cell lung cancer (NSCLC), called MAGRIT, the same MAGE A3 protein vaccine was tested and appeared to have low immunogenicity. There was no clinical response in the treated patients. As a result, this study fell short of its primary goal and was terminated in 2014 (NCT00480025) [40].

Why were these two clinical trials ineffective? First, the eligibility of patients is determined prior to treatment based on MAGE-A3 expression; in such studies, this has often been assessed at the gene level by quantitative RT-PCR, but nothing is known about expression at the protein level or the actual number of MAGE-A3-positive cancer cells. In addition, the formulation of these antigens to stimulate an immune response should be considered. Some researchers attempted to solve the challenges by developing a MAGE-A immunogen as a DNA vaccine with cross-reactivity to the many MAGE-A isoforms, i.e., not simply being limited to MAGE-A3. The MAGE-A DNA vaccine consensus sequence was designed by aligning human MAGE-A (-A2, -A6, -A3, -A12) amino acid sequences. This vaccine’s antitumor efficacy was also tested in vivo in a melanoma model tumor. It was observed that the MAGE-A DNA vaccine triggered a robust immune response, significantly reduced tumor growth, and prolonged survival (Figure 2B) [41].

Other studies used dendritic cell (DC)-based vaccines as vehicles for antigen delivery in different formats. Liyan Lin et al. transduced DCs with a lentiviral vector containing the full-length MAGE-A3 gene [42]. Ramesh B. et al. employed the rAAV-6 capsid mutant vector Y445F to transduce DCs with the MAGE-A3 gene [43]. Aude Bonehill et al. improved the immunostimulatory capacity of dendritic cells by electroporation with an mRNA encoding CD40 ligand, CD70, and a constitutively active Toll-like receptor 4 (TriMix DCs). Additionally, TriMix DCs were also co-electroporated with mRNA encoding MAGE-A3 [44,45]. All of these studies reported that DCs induced MAGE-A3-specific T lymphocytes, which were capable of lysing MAGE-A3-bearing tumor cell lines (Figure 2B).

Furthermore, an RNA lipoplex vaccine encoding for MAGE-A3 and another tumor antigen were developed for a phase I dose escalation study. Three melanoma patients received the vaccine at a low dose. Strong antigen-specific T-cell responses and IFN-a were observed, suggesting that the RNA-LPX vaccine may be suitable for systemic DC targeting. This trail is still ongoing (NCT02410733) (Figure 2B) [46].

However, most vaccine therapies for cancer have focused on cytotoxic T lymphocyte CD8+ (CTL) activation, ignoring the CD4+ helper T cells that play an important role in full CTL activation. To overcome this issue, Norihiko Takahashi and colleagues artificially designed a long hybrid peptide of MAGE-A4, which was composed of MAGE-A4278–299 CD4+ T cell helper epitopes and MAGE-A4143–154 CD8+ T cell killer epitopes (MAGE-A4-H/K-HELP). Initial clinical results showed that the MAGE-A4-H/K-HELP vaccine elicited cellular and humoral responses in one patient with pulmonary metastases of colon cancer. The helper epitope stimulated CD4+ and CD8+ T cell responses, whereas the killer epitope triggered the production of MAGE-A4-specific IgG antibodies. In addition, tumor growth and serum tumor marker levels decreased (UMIN000003489) (Figure 2B) [47].

Recent advances in immune system modulation via the use of antibodies that block immune checkpoints such as the PD1/PD-L1 and CTLA4/B7 axes have paved the way for the optimization of novel human cancer treatment regimens. There is evidence that the combination of an immune checkpoint inhibitor and vaccination improves the survival in patients with metastatic melanoma, representing a milestone [48,49]. 

**Figure 2 cancers-15-01779-f002:**
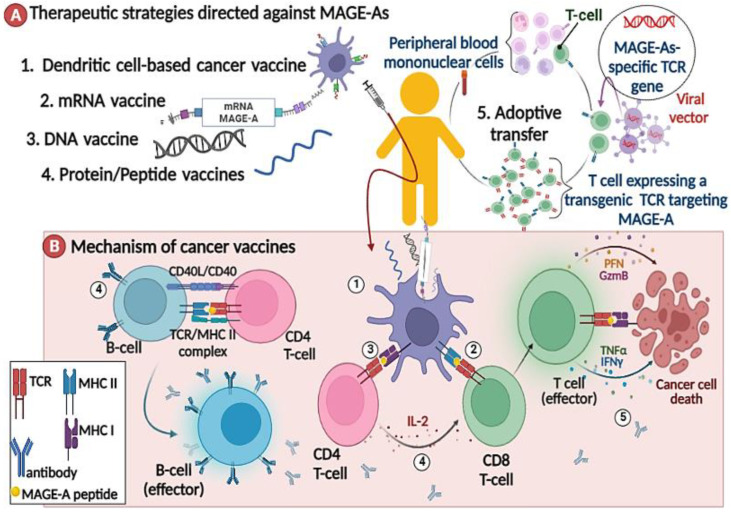
Schematic depiction of therapeutic strategies. (**A**) and their mechanisms (**B**) directed against MAGE-A antigens. (**A**) Many types of MAGE-directed cancer immunotherapy have been developed, including protein or peptide vaccines, DNA or RNA vaccines, cell-based vaccines, and adoptive T cell therapy. (**B**) The immune response to a cancer vaccine consists of several steps: ① Antigen-presenting cells (APC) capture injected MAGE-As, whether DNA, RNA, or peptides, and present them to stimulate CD8 T cells ② via MHC I, and helper CD4 T cells ③ via MHC II. ④ Activated CD4 T cells coordinate immune responses by communicating with other cells and inducing B cells to differentiate into plasma cells. ⑤ Finally, effector T cells, B cells, antibodies, and some cytokines have either a direct or indirect antitumor effect on cancer cells [50].

A completed phase II study in which patients with melanoma stages II or IV were treated with a dendritic cell-based mRNA vaccination encoding several tumor antigens—including MAGE-A3, plus ipilimumab, which is a monoclonal antibody (mAb) directed against CTL4—yielded encouraging results. Trimix-DC-MEL IPI treatment resulted in a long-term favorable outcome and a strong antigen-specific T cell response in patients with a complete response, especially in those patients who evidently made a full recovery after 5+ years of follow-up. This study highlighted combination therapies with a checkpoint inhibitor and a tumor or CT antigen-based vaccine, which may synergize to produce a more effective treatment with long-term clinical remission (NCT01302496) (Figure 3A). 

In another phase I clinical trial, children with relapsed sarcoma and neuroblastoma were administered the demethylating agent decitabine, followed by a DC pulsed with MAGE-A1/A3 and NY-ESO-1 peptides. This combined regimen was tolerated and feasible, as well as triggered an antigen-specific T-cell response in the majority of patients. One patient had a complete response for 3.5 years, and 5 of 10 patients experienced decitabine-related myelosuppression. This was the first study to evaluate the tolerability and potential toxicity of an epigenetic approach in combination with DC-based immunotherapy (NCT01241162) [51].

**Figure 3 cancers-15-01779-f003:**
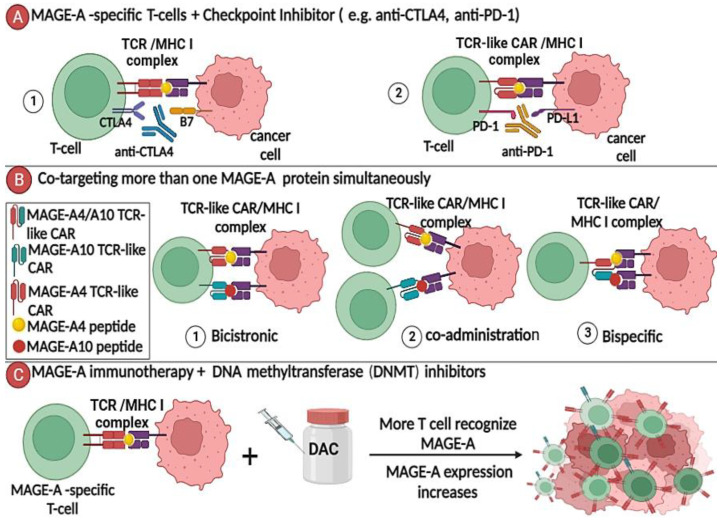
Targeting the MAGE-A antigen in combination with other therapies: proposed approaches. (**A**) TCR/TCR-like CAR-engineered T cells specific for MAGE-As could be combined with immune checkpoint inhibition therapy, disrupting the mechanism that suppresses the immune response to tumor cells, i.e., anti-PD-1 antibodies play a role in blocking the PD-1/PDL1 pathway. (**B**) TCR-like CAR T cells can be used to co-target MAGE-A4 and MAGE-A10, implying the use of vectors and strategies similar to those discussed in a mini-review on multi-targeted CAR T therapy as follows [52]: ① bicistronic vector—can generate one T cell with two TCR-like CARs, one specific for MAGE-A4 and the other specific for MAGE-A10; ② co-administration—producing two distinct cell populations, each with its own TCR, and then redirecting them simultaneously or sequentially; and ③ bispecific vector—can generate one T cell with a single TCR-like CAR specific for MAGE-A4/A10 together. (**C**) Combining DNA methyltransferase inhibitors, such as decitabine (DAC), with vaccination will increase the expression of MAGE-A proteins on cancer surface cells and may generate more MAGE-A-specific T cells and thus reinforce antitumor activity.

#### 2.3.2. MAGE-A-Directed Adoptive T Cell Therapy

Adoptive T cells are one of the most important immunotherapy methods because they use our body’s most powerful weapon in the fight against cancer. Tumor-infiltrating autologous lymphocytes are isolated from the patient, expanded in the laboratory, and then reinfused in increasing numbers back into the patient (Figure 2A). The task of this army of T cells in the body is to surveil, recognize tumor antigen on tumor cells, and then attack the cancer cell. Following lymphodepleting chemotherapy, the adoptive T cell therapy methodology demonstrated objective response rates of more than 50% in patients with metastatic melanoma [53].

In another study, researchers tried to improve adoptive T cell transfer in patients with relapsed, EBV-negative Hodgkin lymphoma (HL) by targeting the MAGE-A4 antigen. The cytotoxic T cells were isolated from HL patients and cultured with dendritic cells that had been pulsed with a library of MAGE-A4 peptides. Prior to treatment, patients were given an epigenetic-modifying drug (decitabine) to investigate its effects on T cells. Decitabine was found to promote MAGE-A4 upregulation in cancer cells and also participate in selective recognition by MAGE-A4-specific T cells, which is consistent with increased antigen stimulation in vivo (Figure 3C). It was concluded that the synergy of an adoptive transfer of MAGE-A4-specific T cells with epigenetic modifying drugs could offer broad prospects for improving the management of relapsed patients [54] and, as such, warrants further clinical investigation.

However, obtaining tumor-infiltrating T cells has proven more difficult with other types of cancer. In recent studies, this approach was extended to a wider group of patients by genetically modifying T cells to generate effector T cells by transduction with retroviral and lentiviral vectors encoding MAGE-A-specific TCRs [55].

In a preclinical study, T cells derived from human peripheral blood mononuclear cells were engineered to express TCR specific for the MAGE-A4 143–151 peptide (NYKRCFPVI) and adoptively transferred to immunodeficient NOG mice following inoculation with MAGE-A4-expressing human tumor cell lines. Next, antigen vaccination was combined with the adoptive transfer of antigen-specific T cells. It was observed that the transferred T cells infiltrated the tumor site and inhibited tumor growth, whereas the combination therapy increased effector T-cell polyfunctionality to produce a detectable antitumor effect [56].

Based on these preliminary findings, a phase I clinical trial was designed to treat patients with MAGE-A4-expressing esophageal cancer by using T cells transduced with retroviruses encoding MAGE-A4143–151 and HLA-A*24:02–specific TCR (Figure 2A). Although the patients did not receive lymphodepletion, the modified T cells could have persisted for more than 5 months in five patients. However, there was discordance between tumor regression and persistence cells. Further, 3/5 patients had a stable disease for more than 27 months. No such T cell responses indicating cross-reactivity to any peptide similarly to the MAGE-A4 peptide were observed. Furthermore, the study highlighted the benefits of preparative lymphodepletion in adoptive T-cell therapy [57]. 

In another study with lymphodepleted patients, researchers noted a concordance between tumor regression and a persistence of transduced T cells using autologous anti-MAGE-A3 112–120/HLA-A*0201 TCR (KVAELVHFL) engineered T cells (NCT01273181). A total of five out of nine patients (seven with metastatic melanoma, one with synovial sarcoma, and one with esophageal cancer) showed a measurable clinical response, and two patients demonstrated ongoing regression 12 months post-treatment. However, three out of nine patients manifested neurotoxicity, and two patients died. The authors concluded that this toxicity was due to the high affinity of these TCRs for the highly homologous MAGE-A12 peptide (KMAELVHFL), which was found to not be readily predictable in terms of being expressed in the CNS [58]. Similarly, two melanoma patients died of cardiogenic shock within a few days of infusion with T cells targeted to the HLA-A*01–restricted MAGE-A3 peptide (EVDPIGHLY). The “off target” toxicity was attributed to the recognition of an unrelated epitope (ESDPIVAQY)-derived protein called titin, which is expressed in the striated muscle (NCT01352286) [59,60].

Another group of researchers isolated two MHC class II-restricted TCRs from melanoma patients after MAGE-A3 vaccination, both of which recognize the same MAGE-A3243-258 peptide (KKLLTQHFVQENYLEY), supporting a previous hypothesis that genetically modified MAGE-3/HLA-DPB1*04:01 CD4+ T cells could lead to cancer patient regression. One TCR represented a regulatory T cell clone, and one represented an effector clone. The MAGE-A3 TCR, which was originally derived from a clone with a regulatory T cell phenotype, showed a higher affinity than the effector TCR and was then selected for clinical trials [61]. The safety and efficacy of this regulatory TCR were evaluated in a phase I dose escalation study in patients with various types of cancer. One patient with metastatic cervical cancer, who received 2.7 × 10^9^ cells, achieved a complete objective response (lasting 29 months). Three of the nine patients treated at the maximum dose experienced objective partial responses, as per the following: one patient with esophageal cancer; one with osteosarcoma (duration: 4 months); and one with urothelial cancer (duration: 19 months). This study provided new insights into adoptive therapy by using genetically engineered CD4+ T cells to express an MHC class II-restricted antitumor TCR that targets MAGE-A3 (NCT02111850) [62].

In two recent studies, patients with advanced NSCLC, head and neck squamous cell carcinoma, melanoma, or urothelial carcinoma were treated with genetically engineered autologous T cells that express a high-affinity MAGE-A10-specific TCR targeting MAGE-A10-positive tumors in the context of HLA-A*02 (ADP-A2M10). The best response included stabilization of the disease in four patients with no evidence of any toxicity that was related to off-target binding (NCT02592577; NCT02989064) [63,64]. However, due to the overlapping expression of MAGE-A10 and MAGE-A4 in tumors, the ADP-A2M10 experiments were subsequently stopped. Instead, a study (NCT03132922) targeting the MAGE-A4 antigen with ADP-A2M4 was initiated [65], and favorable responses were observed in patients with synovial sarcoma, non-small cell lung cancer, and head and neck cancer [66].

Many trials with ADP-A2M4 targeting the MAGE-A4 antigen are currently ongoing (NCT03132922, NCT04044768, and NCT04044859). The co-expression of MAGE-A4 and MAGE-A10 in several tumors suggests that future research should focus on co-targeting multiple MAGE-A antigens (Figure 3B).

In addition to genetically modified T cells with transgenic TCR, chimeric immunoglobulin (Ig)-based receptors with restricted MHC specificity have also been successfully generated in gene-based strategies. Willemsen et al. published the first preclinical report, in which they generated T cells expressing a Fab-based chimeric receptor that specifically recognizes a MAGE-A1160–169-derived peptide in the context of HLA-A1 (EADPTGHSY). These chimeric antigen receptor T cells (CAR T cells) specifically respond to and induce the cytolysis of melanoma cells through the production of TNF-α and IFN-γ. This study paved the way for the peptide/MHC-specific Fab fragment-based CAR T cell as a novel alternative to engineered TCRs, which could be a promising tool in immunotherapy (Figure 3A) [67]. CAR T cells are now the dominant approach used for adoptive T cell therapy in cancer [68].

All the preclinical and clinical studies on MAGE-As discussed in this review are summarized in Table 1.

## 3. Future Perspective and Conclusions

There is a considerable body of published data suggesting that strategies directed against MAGE-As may provide rational routes to combat MAGE-A-expressing tumors. MAGE-A proteins, as discussed in this review, exhibit remarkable properties for promoting oncogenicity and metastasis, which is a significant criterion for the selection of MAGE-A antigens for future immunotherapy. Advances in our understanding of the tumor microenvironment, the underlying immune evasion mechanisms of tumors, and the safety considerations arising from clinical studies all inform the pursuit of an ideal therapy. Given our current knowledge, preclinical and clinical trials with different approaches directed against MAGE-As have been able to induce heterogeneous immune responses, rarely causing “off-target” toxicity, which indicates that the MAGE-As are not solely expressed in germ cells but are also expressed in cells of multiple tumors. As a result, investigating the expression of the target antigen for the patients’ eligibility as a prerequisite for treatment may be worthwhile.

Moreover, to rigorously assess new MAGE-A targets that can be utilized in immunotherapy, further tests are needed to verify specificity issues, such as screening platforms for the detection of cross-reactive epitopes that are shared by candidate MAGE-A immunogens with host proteins. Crystallography studies can also be applied to identify the specific MAGE-A peptide residues that bind to TCR or CAR.

Furthermore, strategies can be combined to optimize novel treatments and refine the manner in which these treatment regimens impact the immune system. Regarding this, mAbs that block immunological checkpoints have sparked significant interest, and the main working mechanism of these mAbs is to disengage the brakes of impeding T cells that are specific for tumor antigens. For instance, anti-PD-1 and anti-CTLA-4 therapeutic mAbs have achieved impressive success and have also been approved by the FDA for the treatment of certain cancers [69]. Thus, immune checkpoint inhibition should be considered in conjunction with conventional immunotherapy for MAGE-A antigens to improve outcomes. 

The combination of chemotherapy with MAGE-A-related immunotherapy to eliminate MAGE-A expressing tumor cells may also prove synergistic. Additionally, as previously stated, treatment with DNMT inhibitors could enhance the upregulation of MAGE-A antigens. Similarly, combining demethylation of MAGE-A antigens with vaccination or adoptive T-cell transfer may generate more MAGE-A-specific T cells and reinforce antitumor activity.

Furthermore, given the homologous structure and heterogeneous expression of MAGE-A antigens across many tumor types, an epitope common to all MAGE-A antigens could be a potential candidate for broad-spectrum tumor immunotherapy. 

The comprehensive understanding of the crosstalk between cancer and immune control, the encouraging clinical data in the framework of combinatorial therapies, and the ongoing clinical trials directed against MAGE antigens—all provide hope and further insight into the birth of a new era in cancer immunotherapy.

## Figures and Tables

**Figure 1 cancers-15-01779-f001:**
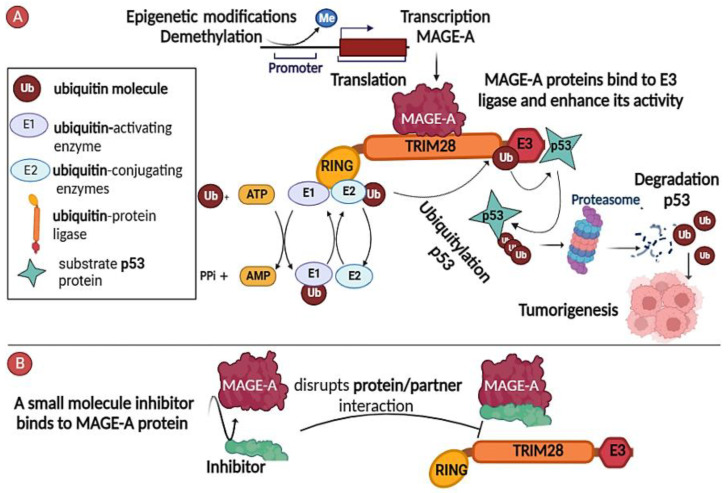
MAGE-A antigens as cancer therapeutic targets. (**A**) MAGE-A plays a role as the scaffold for the ubiquitin-protein ligase E3, which forms a complex with the TRIM28 protein. The MAGE-A genes are activated by epigenetic modifications, such as demethylation. MAGE-A binds to TRIM28, enhancing the ubiquitin ligase E3, which binds the tumor suppressor p53 and tags it for degradation by the proteasome, thus resulting in tumorigenesis. (**B**) Small molecule inhibitors of the MAGE-A interaction with partner protein TRIM28 are being developed to exploit MAGE-A proteins as cancer therapeutic targets.

**Table 1 cancers-15-01779-t001:** An overview of MAGE-A-related preclinical and clinical trials.

Identifier	Phase	Type of Tumor	MAGE-A	HLA	Formulation of Therapy	Results
NCT00086866 [37]	II	Melanoma	MAGE-A3		Recombinant protein	Well tolerated, Three complete responses, Humoral response and cellular response against MAGE-A3
NCT00849875 [39]	I/II				Peptide vaccine + Dacarbazine	One complete response, Three partial responses, Humoral response and cellular response against MAGE-A3
PMC6319943 [41]	In vivo	Melanoma model	Consensus sequence shared of (MAGE-A1, A2, A3, A5, A6, and A8)		DNA vaccine	Robust immune response, Slowed tumor growth, Prolonged survival
PMID: 24322180 [42]	In vitro	Melanoma	MAGE-A3-transduced dendritic cells by lentiviral vectors (rLV/MAGE-A3)		Dendritic cell (DC)	Induced MAGE-A3-specific T lymphocytes, T lymphocytes showed a significant lysis activity against MAGE-A3-bearing tumor cell lines
PMID: 24406390 [43]			MAGE-A3-transduced dendritic cells by rAAV-6 capsid mutant vector Y445F (rAAV-6-MAGE-A3)			Produced CTLs, CTLs lysed of epithelial ovarian cancer cells
PMID: 19417017 [45]	I		Co-electroporate-Trimix Dcs with mRNA encoding MAGE-A3		TriMix-DC vaccine	Elicited antigen-specific T-cell responses through vaccination
NCT02410733 [46]	I		MAGE-A3 plus NY-ESO1		RNA-LPX vaccine	Induced IFNα and strong antigen-specific T-cell responses, One complete response
PMID: 22221328 [47]	I		Helper/killer-hybrid epitope long peptide MAGE-A4		Peptide vaccine	CD4+ and CD8+ T cellular responses specific for MAGE-A4, MAGE-A4-specific IgG antibodies
NCT01302496 [49]	II		Co-electroporated-TriMix DCs with mRNA encoding MAGE-A3 and other TAA		DCs vaccine + Ipilimumab (anti-CTLA-4)	TAA-specific robust immune response, Some patients had a durable complete remission of metastatic melanoma.
NCT01241162 [51]	I	Sarcoma Neuroblastoma	MAGE-A1, MAGE-A3, and NY-ESO-1-pulsed DCs		DCs vaccine + Decitabine (DAC)	One complete response, 5/10 patients experienced DAC-related myelosuppression
PMC3218253 [51]	In vitro		MAGEA-4-pulsed DCs co-cultured with autologous T -cells		Preparing of autologous MAGE-A4 specific T- cells (in vitro) + Decitabine vaccine in patients	Generated MAGE-A4-specific cytotoxic T-cell in vitro, Decitabine-expanded MAGE-A4-specific T cells in patients.
PMID: 21951605 [56]	In vivo	Model tumor NOG mice injected with the cell line QG56 of human lung cancer (MAGE-A4 + HLA-A*2402−) or KE4 of human esophageal cancer (MAGE-A4 + HLA-A*2402+)	MAGE-A4 143–151 peptide (NYKRCFPVI)	HLA-A*2402	Adoptive transfer (TCR- modified T cells) + Peptide vaccine	Generated effector T-cell polyfunctionality, CTLs had antitumor effectivity
PMID: 25855804 [57]	I	Esophageal cancer			Adoptive transfer (TCR-modified T cells)	3/5 patients had a stable disease for more than 27 months, T cells persisted for more than 5 months in 5 patients, Discordance between tumor regression and persistence cells
NCT01273181 [58]	I//II	Melanoma Synovial sarcoma Esophageal cancer	MAGE-A3 112–120 (KVAELVHFL)	HLA-A*0201	Adoptive transfer (TCR-modified T cells) + chemotherapy	Two patients had an on-going regression 12 months post-treatment, Two patients died due to neurotoxicity, The research was closed
NCT01352286 [60]		Melanoma	MAGE-A3 peptide (EVDPIGHLY)	HLA-A*01		Two patients died due to cardiogenic shock, The research was closed
NCT02111850 [62]		Cervical cancer Esophageal cancer Urothelial cancer Osteosarcoma	MAGE-A3 243–258 peptide (KKLLTQHFVQENYLEY)	HLA-DPB1*04:01		One complete objective response (lasting 29 months), Three objective partial responses (lasting 4–19 months)
NCT02592577 NCT02989064 [64,66]		NSCLC Head and neck squamous cell carcinoma	MAGE-A10	HLA-A*02		The best response: four patients had stable disease and no evidence of toxicity, The research was closed
NCT03132922 [66]	I	Synovial sarcoma Non-small cell lung cancer Head and neck cancer	MAGE-A4	HLA-A2	Adoptive transfer (TCR-modified T cells) + Low-dose radiation	The study is still ongoing
PMID: 11894998 [67]	In vitro	Melanoma cells	MAGE-A1160–169	HLA-A1	Fab-based chimeric receptor, specific for MAGE-A1/HLA-A1	Induced cytolysis of melanoma cells through the production of TNF-α and IFN-γ

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
