# Peer review of "The Melanoma-Associated Antigen Family A (MAGE-A): A Promising Target for Cancer Immunotherapy?"

_cancers, 2023, doi:10.3390/cancers15061779_

Round 1

Reviewer 1 Report

In this Review, Alsalloum et al provided a summary of the biological and mechanistic insights of melanoma-associated antigens. They also discussed the potential of using melanoma-associated antigens as therapeutic options to treat cancer patients. They concluded by giving a broad perspective of targeting MAGE-A family for cancer treatment. However, the manuscript lacks clarity due to grammatical errors and figures are not straightforward and clear to understand.

Specific comments:

1)Figure 1 is very confusing with (a) and (b) titles. Suggestion to remove headers in the figures and elaborate in the figure caption.

2)Figure 2 is not clear as the font and different cellular icons and symbols and fonts are overlapped. 

3) confidence measures This manuscript requires extensive English and format corrections. This include but not limited to: (i) Reference numbers are cited after the statement; (ii) “Sponsors” is not a correct term for this statement “Cancer vaccines are a type of active immunotherapy that sponsors the patient’s im-158 mune system in its battle against cancer. ”; (iii) (Figure 2c_2); (iv) format issue for citation: Table 1. A list of MAGE-A-related preclinical and clinical trials that have been registered with re-363 spective websites (https://pubmed.ncbi.nlm.nih.gov/ and https://clinicaltrials.gov). (v) All of these preclinical and clinical studies on MAGE-As are summarized in (Table1).

Author Response

Dear reviewer 1, We sincerely appreciate your valuable comments and suggestions on our manuscript that helped us to improve the quality of the work. We revised the manuscript according to your comments. Revised parts are marked up using the “Track Changes” function.

Point 1: In this Review, Alsalloum et al provided a summary of the biological and mechanistic insights of melanoma-associated antigens. They also discussed the potential of using melanoma-associated antigens as therapeutic options to treat cancer patients. They concluded by giving a broad perspective of targeting MAGE-A family for cancer treatment. However, the manuscript lacks clarity due to grammatical errors and figures are not straightforward and clear to understand.

Response 1: To improve the English and readability of our manuscript, it has undergone English language editing by MDPI. “The text has been checked for correct use of grammar and common technical terms and edited to a level suitable for reporting research in a scholarly journal.” MDPI employs experienced, native English-speaking editors. We've attached a certificate of English editing.

Point 2: Figure 1 is very confusing with (a) and (b) titles. Suggestion to remove headers in the figures and elaborate in the figure caption. And Figure 2 is not clear as the font and different cellular icons and symbols and fonts are overlapped.”

Response 2: We agree that Figures 1, 2 seemed to be confusing.

  • We removed headers in Figure 1 and elaborated in the figure caption.
  • We divided Figure 2 into two separate figures with a consistent font and icons, and we edited the legends to add missing information.

Point 3: Confidence measures This manuscript requires extensive English and format corrections. This include but not limited to: (i) Reference numbers are cited after the statement; (ii) “Sponsors” is not a correct term for this statement “Cancer vaccines are a type of active immunotherapy that sponsors the patient’s immune system in its battle against cancer.”; (iii) (Figure 2c_2); (iv) format issue for citation: Table 1. A list of MAGE-A-related preclinical and clinical trials that have been registered with respective websites (https://pubmed.ncbi.nlm.nih.gov/ and https://clinicaltrials.gov). (v) All of these preclinical and clinical studies on MAGE-As are summarized in (Table1).”

Response 3: All of confidence measures have been addressed in the manuscript.

  1. Reference numbers were cited after the statement;
  2. “sponsorswas edited to “supports”
  • (Figure 2c_2): This type of citation was removed, and all citations were edited to, for example, (Figure 1a).
  1. We also noticed that the table's title appeared to be ambiguous. The table summarizes the preclinical and clinical studies on MAGE-A antigens discussed in this review. The title has been changed to "Table 1. An overview of MAGE-A-related preclinical and clinical trials." The references for each clinical trial were considered separately in the table.
  2. All the preclinical and clinical studies on MAGE-As discussed in this review are summarized in Table 1.

Reviewer 2 Report

The scientific content of this manuscript provides an appropriate review on the topic of MAGE-A as a target for cancer immunotherapeutics.  However, this manuscript is riddled with errors in grammar, sentence syntax, and word usage, along with rambling and sometimes incohesive and/or redundant statements/discussions of different MAGE-A-related topics.  Errors in word usage or syntax in some cases resulted in imprecise statements that were unintentionally inaccurate.  The manuscript requires a significant revision for English language and a "clean up" or editing of the text to generate a more fluid discussion of this topic.  Some specific and some general comments below could be used by the authors to produce a manuscript revision suitable for subsequent review and consideration for publication.

 Comments for editing:

Line 25:  "progressing surviving cancer cells" edit to "cancer cells with an increasing potential for survival"

Line 27:  "meaning" edit to "suggesting"

Line 41:  "in the blood" edit to "in the blood, lymphoid tissues and at effector sites"

Line 50:  "immune system. For example, by concealing cancer antigens or secreting immune-suppressing molecules " edit to "immune system, including concealment of cancer antigens or secretion of immune-suppressing molecules".

Line 59:  "which made it a major step" edit to "which have been major steps"

Lines 66-69:  "Cancer/testis (CT) antigens, of which tumor antigens have been identified with normal expression only in testicular, ovarian, fetal, and placental germ cells, and therefore they were dubbed cancer/testis antigens (CTAs).[4,5] but are also expressed in malignancies.[6,7]"  These statements include incomplete sentences and redundancies. 

Edit to "Cancer/testis (CT) antigens match these criteria for an ideal target as CT antigens are tumor antigens identified with normal expression restricted to testicular, ovarian, fetal, and placental germ cells, but may also be detected in malignancies [6,7]."

Lines 69-70:  "Cancer/testis (CT) antigens are an immune-privileged site" edit to "CT antigens are normally located in an immune-privileged site".

Line 73:  "cancer testis antigen" edit to "CT antigen".  All subsequent text containing "cancer testis antigen" should consistently use the term CT antigen.

Lines 74-75: "recognizing autologous cytotoxic T cells of the melanoma cell line in vitro" edit to "recognition of a melanoma cell line in vitro by autologous cytotoxic CD8 T cells."

Line 83:  "winged helix motif. It was assumed that this region plays" edit to "winged helix motif that was assumed to play" (two sentences combined to one).  The citation for this statement is out-dated.  An additional but very general (as you discuss in more detail in next section) summary statement regarding what is currently known and reported (and not "assumed") for the activities of the MHD from more recent (2020 and later) reviews, should be the final sentence in this paragraph.

Lines 85-88:  "This homology between MAGE antigens (MAGEs) of individual family members has made it difficult for researchers to develop specific antibodies to analyze their expression, and research teams have overcome this problem by measuring the mRNA levels of specific MAGEs.[13]" This statement is confusing, not critical to the discussion of MAGE-A and should be deleted.  Ref. 13 is also not an appropriate reference for this statement.

Line 101:  "can bind to the p53 core and recruit it, down-regulating" edit to "can bind to the p53 core and down-regulate"

Line 110:  "They found" edit to "It was shown".

Line 113-116:  "This explains why MAGE- As are not normally expressed in somatic tissues due to DNA hypermethylation of CpG dinucleotides in promoters, which prevents transcription factors such as Ets and SP1 from binding.[26,27]"  This sentence is too rambling and needs re-wording for clarity.

Line 117:  "has made them a unique target of efficacious treatment for a long time" edit to "these CT antigens have received continued interest and investigation over time as worthy targets of therapeutic cancer strategies."

Line 127:  "MAGE-As" edit to "MAGE-A family member proteins."

Line 134: "ligase); they screened a library" edit to "ligase) based on screening of a library"

Line 136: "They reported" edit to "Results of the screen revealed".  " They" is often and inappropriately used in sentences in this manuscript to refer to specific terms such as "investigators" or "results" etc.

Line145:  "in particular" edit to "selectively" for a better word choice.

Lines 149-150:  Use of the term "neoantigens" for MAGE-A CT antigens may not be appropriate as neoantigens are often defined as peptides that arise from mutations due to DNA damage within the tumor, which is not true for MAGE-A CT antigens.  The use of "universal antigen" might be considered instead for MAGE-As, particularly as their expression is detected in multiple types of cancers.

Line 155:  "immune infiltrates" edit to "immune cell infiltrates"

Line 165:  Please be more precise in the description of the MAGE-A3 protein vaccine.  Was the immunogen a recombinant protein or a peptide vaccine (as described in Table 1)?

Line 166:  "measure the immune response" should be edited to "to induce tumor-specific immune responses".

Line 168:  The "group treated with AS15 immunostimulant" is assumed to be the group treated with a combination of MAGE-A3 protein vaccine and AS15.  If so, the statement should read "treated with MAGE-A3 protein combined with AS15 immunostimulant". 

Line 173:  "has not been" edit to "was not".

Line 180:  "used" edit to "tested".

Line 182:  Please define "its main goal".  Are you intending to say that no clinical response was observed in treated patients- hence the study failed to demonstrate any clinical response/efficacy?

Line 193-194:  Please be more specific than "tumor's thriving".  Was tumor size, primary and/or metastatic lesions, reduced or eliminated?

Line 196:  "manners" edit to "formats".

Line 205:  "in the" edit to "for a".

Line 206:  "IFN" should be edited to "IFN-a" based on information in Table 1.

Line 208:  "research" should be edited to "trial".

Line 212:  "helper" should also include "CD4+ T cell helper".

Line 213:  "killer" should also include "CD8+ T cell killer".

Line 216:  "CD8+ T cells" edit to "CD8+ T cell responses".

Line 220:  "PD1" edit to "the PD1/PD-L1 axis".

Lines 221-223:  The statement, "There is milestone evidence that the combination of an immune checkpoint inhibitor and vaccination improves survival in patients with metastatic melanoma. [45]" is cited incorrectly as the cited reference (45) relates to an early trial testing only ipilimumab (CTLA-4 blocking antibody).  This statement should be cited with reference #46 and should serve as the opening statement for the following paragraph (lines 245-253) discussing the trial (showing "milestone evidence") testing ipilimumab in combination with a cancer vaccine that includes MAGE-A as an immunogen.

Line 248:  "a long-term outcome" edit to "a long-term favorable outcome".

Line 251:  "with tumor or testis antigens" edit to "with a checkpoint inhibitor and tumor or CT antigen-based vaccine".

Line 252:  "indicating that the synergy of these regimens would be more effective to achieve successful treatment" edit to "may synergize to produce a more effective treatment".

Line 256:  "pulses" edit to "pulsed".

Line 258:  Please define or write out the phrase for "DAC".

Line 267:  "reveal tumor antigen on the cells" edit to "recognize tumor antigen on tumor cells".

Line 285:  "MAGE-As-specific" edit to  "MAGE-A-specific".

Lines 283-293:  Combine these two paragraphs into one paragraph.

Line 284:  "T cells. Effector T cells are now being transduced with" edit to "T cells to generate effector T cells by transduction with".

Line 289:  "Then, the antigen" edit to "Next, an antigen"  

Line 291:  Replace "frustrated its" with "inhibited tumor".

Line 292:  "making the antitumor effect visible" edit to "to produce a detectable antitumor effect".

Line 295-296:  "using transduced T cells using retroviruses" edit to "using T cells transduced with retroviruses".

Lines 296-299:  Although the patients didn’t receive lymphodepletion, T-cells could have persisted for more than 5 months in 5 patients. However, there was discordance between tumor regression and persistence cells."  These statements are unclear in meaning and need re-wording.  "could have persisted" is vague and unclear as written.  Does "persistence cells" relate to "persistence of transduced T cells"??  

Lines 299-300:  Does "No cross-reactions to" refer to "No T cell responses cross-reactive to"?

Line 302:  Again does "persistence cells" refer to "persistence of transduced T cells"?

Line 305:  "2 of them had" edit to 2 patients demonstrated".

Line 306:  "2 of them" edit to "2 patients".

Lines 307-309:  "The authors elucidated this toxicity due to the high affinity of these TCRs for the highly homologous MAGE-A12 peptide (KMAELVHFL), which was found to not be readily predictable to be expressed in the CNS." This sentence is convoluted with incorrect word usage.  Did you mean to communicate:  "The authors concluded (or suggested) that this toxicity was due to the high affinity of these TCRs for a MAGE-A12 peptide (KMAELVHFL), that was highly homologous to an epitope previously uncharacterized for expression in CNS tissue." ?

Lines 314-320:  This paragraph is poorly written and could be deleted as the reference and topic is not that important to this section.

Line 325:  "One TCR was from a regulatory" edit to "One TCR represented a regulatory"; "one was from an effector" edit to "one represented an effector".

Line 327:  "the other TCR" edit to "the effector TCR"

Line 328:  "this TCR were" edit to "this regulatory TCR were".

Line 330:  "109" edit to 109.

Line 340:  "patients. no evidence" edit to  "patients with no evidence".

Line 342:  "started" edit to "was initiated".

Lines 342-343:  "considerable" might be replaced with "significant" or "favorable".

Line 345-346:  "Adaptimmune data, in which the expression of MAGE-A10 and MAGE-A4 overlapped" is vague without enough detail or description for the reader to follow, and there is no reference cited.  Do you mean to say " Adaptimmune data which revealed overlap of expression for MAGE-A10 and MAGE-A4 CT antigens for certain tumors"? 

Line 347:  "was higher" edit to "was shown to be higher". 

Lines 356-357:  "CAR T cells" edit to "chimeric antigen receptor T cells (CAR T cells)"  

Line 359:  "Fab fragment as" edit to "Fab fragment-based CAR T cell as" and "to TCR" edit to "engineered TCRs".

Lines 359-360:  "immuntherapy [62] and then this approach has grown and is now called TCR like CAR" edit to  "immunotherapy [62].  CAR T cells are now the dominant approach used for adoptive T cell therapy in cancer."

Line 367:  "Although this is just a start, there is already published data suggesting that strategies" edit to "There is a considerable body of published data suggesting that strategies"

Lines 369-370:  "characteristics for" edit to "properties for promoting".

Line 373:  "all inspire" edit to "all inform".

Line 377:  "germ cells" edit to "germ cells but are also expressed in cells of multiple tumors."

Line 379:  "for the vigor of" edit to "to rigorously assess".

Lines 380-381:  "such as screening the binding to an antigen with a similar sequence to the target antigen." edit to "such as screening platforms for detection of cross-reactive epitopes shared by candidate MAGE-A immunogens with host proteins."

Line 383:  "Further" edit to "Furthermore".

Line 385:  "a lot of" edit to "significant".

Line 385:  "interest. The main mechanism of mAbs is" edit to "interest and the main working mechanism of these mAbs is"

Line 392:  "may also function synergistically" edit to "may also prove synergistic".

Lines 383-392:  Combine these two paragraphs into one paragraph.

Lines 393-397:  Condense this paragraph with the following paragraph (lines 398-401) to state that "Combination of chemotherapy with MAGE-A-related immunotherapy to eliminate MAGE-A expressing tumor cells may also prove synergistic.  Also, as previously stated, treatment with DNMT inhibitors could enhance the upregulation of MAGE-A antigens.  Similarly, combining demethylation of MAGEA antigens with 399 vaccination or adoptive T-cell transfer may generate more MAGE-A-specific T cells and 400 reinforce antitumor activity."

Line 407:  "all of this provides" edit to "all provide".

Author Response

Dear reviewer 2, We would like to thank you for taking the necessary time and effort to revise the manuscript. We sincerely appreciate all the valuable comments and suggestions, which helped us improve the quality of the work. All of the recommendations have been addressed in the manuscript. Revised parts are marked up using the "Track Changes" function.

Point 1: The scientific content of this manuscript provides an appropriate review on the topic of MAGE-A as a target for cancer immunotherapeutic.  However, this manuscript is riddled with errors in grammar, sentence syntax, and word usage, along with rambling and sometimes incohesive and/or redundant statements/discussions of different MAGE-A-related topics.  Errors in word usage or syntax in some cases resulted in imprecise statements that were unintentionally inaccurate.  The manuscript requires a significant revision for English language and a "clean up" or editing of the text to generate a more fluid discussion of this topic.  Some specific and some general comments below could be used by the authors to produce a manuscript revision suitable for subsequent review and consideration for publication.

Response 1: We revised the manuscript in response to insightful comments and suggestions, and it has also undergone English language editing by MDPI to improve the English and readability. "The text has been checked for correct grammar and common technical terms and edited to a level suitable for reporting research in a scholarly journal." MDPI employs experienced, native English-speaking editors. We've attached a certificate of English editing.

Round 2

Reviewer 1 Report

Hi Authors, 

Can you please make sure the figures are of publication quality where the fonts are still small, and some are overlaps and labels (e.g. A, B, C) are in symbols.

Thank you.

Author Response

We would like to thank you for taking the necessary time and effort to revise the manuscript and sincerely appreciate your valuable comments . All of the recommendations have been addressed in the manuscript. We modified the font size and the symbols .we used (PNG) File Format 600 DPI .

Reviewer 2 Report

The revised manuscript has addressed all the significant errors in English language syntax, word usage and spelling observed in the previous version of this manuscript, to generate a manuscript that is highly informative for the topic of MAGE-A as a target for immunotherapeutic approaches in cancer therapy and is acceptable for publication.

Author Response

We sincerely your positive comments on our manuscript. And we would like to thank you for taking the necessary time and effort to revise the manuscript.